# Estimating Differential Equations from Temporal Point Processes

**Shuichi Miyazawa**                                                      *miyazawa@ism.ac.jp*
*Department of Statistical Science,*
*The Graduate University for Advanced Studies*

**Daichi Mochihashi**                                                      *daichi@ism.ac.jp*
*The Institute of Statistical Mathematics*

**Reviewed on OpenReview:** *https://openreview.net//forum?id=cJgHzw8Qhq*

## Abstract

Ordinary differential equations (ODEs) allow interpretation of phenomena in various scientific fields. They have mostly been applied to numerical data observed at regular intervals, but not to irregularly observed discrete events, also known as point processes. In this study, we introduce an ODE modeling of such events by combining ODEs with log-Gaussian Cox processes (Møller et al., 1998). In the experiments with different types of ODEs regarding infectious disease, predator-prey interaction, and competition among participants, our method outperformed existing baseline methods assuming regularly observed continuous data with respect to the accuracy of recovering the latent parameters of ODEs. Through both synthetic and actual examples, we also showed the ability of our method to extrapolate, model latent events that cannot be observed, and offer interpretability of phenomena from the viewpoint of the estimated parameters of ODE.

## 1 Introduction

Ordinary differential equations (ODEs) have been used in various scientific fields to model naturally occurring phenomena (Jones et al., 2009; Simmons, 2016). ODEs can be derived from first principles or mathematical models developed by a domain expert to explain phenomena (Murray & Murray, 2003; Brauer, 2017). Using domain knowledge of the phenomena, one can specify the form of the ODE. However, the parameters of the ODE are often unknown. Since these parameters can help understand phenomena, the inverse problem, *i.e.*, estimation of the parameters from observed data, has been studied as one of the main problems in the scientific literature (Kryazhimskiy & Osipov, 1995). Primary data for this purpose have been numerical observations that are regularly collected; however, in practice, the data are often not collected under such experimental control, limiting the application of ODE modeling. To broaden the application scope of ODEs, here we focus on event data, which are discrete phenomena that occur irregularly in continuous time, also known as discrete events. This type of data has previously not been the subject of ODE modeling. Thus, we address the problem of estimating the parameters of ODEs from such discrete data.

Event data are ubiquitous in various domains, from science to industry, *e.g.*, earthquakes, blog posts, tweet posts, and business transactions. The properties of event data are significantly different from those of numerical data commonly used for ODE modeling: event data only consist of the time of their occurrences, and while they may include some metadata, they do not come with numerical information by themselves. More importantly, event data are often observed irregularly, leading to the application of existing ODE modeling methods that assume regularly observed data to be a non-trivial problem.

To model such discrete events, temporal point processes (TPPs) have been studied extensively (Daley & Vere-Jones, 2007; Yan, 2019). TPPs model the probability of such event occurrences using a latent intensity function; in particular, a model where the intensity varies over time is called a Cox process (Cox, 1955). To

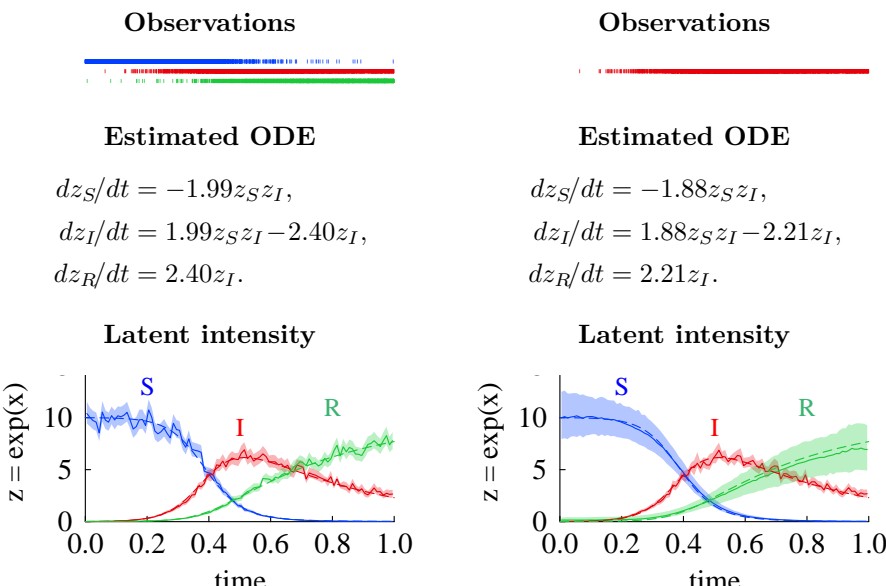

Figure 1: Illustration of the proposed method using SIR equations in epidemiology as an ODE. The Parameters of ODE and latent intensities are estimated from discrete events (point processes) for two cases: when all types of observations are available (left column), and when only type I observations are available (right column). The shaded areas represent the 75% quantiles of the estimated intensities.

estimate the parameters from discrete events, this study defines an ODE-guided Poisson process, where the latent intensity is assumed to follow the dynamics of an ODE. An illustrative result of ODE-guided Poisson process is shown in Fig. 1. To infer this model from discrete events, we propose an inference method that combines the log-Gaussian Cox process (LGCP) (Møller et al., 1998) with the fast Gaussian process-based gradient matching (Wenk et al., 2019), which is an efficient ODE parameter estimation method. Because the posterior probability density distribution of the proposed model is not given in a closed form, an approximate inference in Bayesian framework is necessary. We used the Markov Chain Monte Carlo (MCMC) method, an algorithm widely used in approximate posterior inference, to obtain samples of latent variables from complex, high-dimensional probability distributions (Gelman et al., 1995; MacKay, 2003).

Recently, Neural ODEs (Chen et al., 2018) have been proposed for flexible time series modeling using deep neural networks with ODEs. Subsequently, extensions of Neural ODE incorporating event modeling have been proposed (Jia & Benson, 2019; Rubanova et al., 2019; Chen et al., 2020). Although these studies aim to model discrete events as well as this study, they employ implicit ODEs that do not have an explicit form, thus precluding the interpretability of the data from their parameters.

In contrast, the parameters of explicit ODEs allow the interpretation of phenomena related to event data, which is the aim of this study. Most existing ODE parameter estimation methods assume a discrete time model with regularly observed continuous data; however, TPPs are continuous time models that handle irregularly observed discrete events. The main issue is to fill such gap between the parameter estimation of ODEs and TPPs, providing new flexibility and applicability in ODE modeling. In this study, we

- define an ODE-guided Poisson process, which is a generative model of discrete event sequences governed by an explicit ODE,

- develop an approximate inference method for ODE-guided Poisson process by incorporating LGCP with ODE parameter estimation method,

- firstly demonstrate on synthetic data simulated by using ODEs regarding infectious disease, predator-prey interaction, and competition, that our method outperforms baseline ODE parameter estimation methods developed for regularly observed numerical data with respect to the accuracy of estimating parameters of ODE, showing the effectiveness of combining TPPs with ODE parameter estimation,

- show that our method improves extrapolation performance by introducing mechanistic knowledge through ODEs and can be used even when some types of events are unavailable as shown in the right column of Fig. 1,
- apply the proposed method to real-world data to present examples of modeling the generation process of event data with ODEs, and interpreting phenomena from the viewpoint of estimated ODE parameters.

## 2 Backgrounds

### 2.1 Modeling with Differential Equations

Consider a dynamical system of ODE consisting of $K$ component states. In general, the state of the system at time $t \in \mathbb{R}$ can be represented by a vector $\mathbf{z}(t) = \{z_k(t)\}_{k=1}^K$, where $z_k(t)$ denotes the state quantity of the $k$-th component. Assuming that the observations $\mathbf{y}(t) = \{y_k(t)\}_{k=1}^K$ are available for $N$ time points $\boldsymbol{t} = t_1, \ldots, t_N$ (regularly spaced), the relationship between $\mathbf{y}(t)$ and $\mathbf{z}(t)$ can be described with the additive i.i.d. Gaussian noise $\boldsymbol{\epsilon}(t) \sim \mathcal{N}(\mathbf{0}, \sigma^2 \boldsymbol{I})$, as follows:

$$\mathbf{y}(t_i) = \mathbf{z}(t_i) + \boldsymbol{\epsilon}(t_i), \quad i = 1, \ldots, N. \tag{1}$$

Given $\mathbf{z}(t)$ and ODE parameters $\boldsymbol{\theta}$, the derivatives of the states are computed over a known function $\mathbf{f}(\cdot)$ as

$$\partial \mathbf{z}(t)/\partial t = \mathbf{f}(\mathbf{z}(t), \boldsymbol{\theta}). \tag{2}$$

In the inverse problem for ODEs, we need to estimate $\boldsymbol{\theta}$ given observations $\mathbf{y}$; however, this is challenging for a general nonlinear function $\mathbf{f}(\cdot)$ due to the lack of a closed form solution.

In this study, we focus on ODEs with non-negative states, $\mathbf{z}$. Generally, in ODEs, it is not guaranteed that the state variables are non-negative, and whether they are non-negative depends on the phenomenon being modeled and the problem setting. For example, in physics, the states of harmonic oscillators can have negative values. However, in mathematical models of population dynamics or infectious diseases, it is natural to define states as non-negative. In the following, we introduce examples of ODEs addressed in this study.

**SIR equations.** The SIR (susceptible, infected, and recovered) equations in epidemiology describe the spread of infectious diseases (Kermack & McKendrick, 1927). This system handles three types of states $z_S$, $z_I$, and $z_R$, which denote the sizes of susceptible, infected, and recovered populations, respectively. Using $\{a, b\}$ as the parameters of ODE, the equations are expressed as follows:

$$\frac{dz_S}{dt} = -a z_S z_I, \ \frac{dz_I}{dt} = a z_S z_I - b z_I, \ \frac{dz_R}{dt} = b z_I.$$

**Predator-prey equations.** In mathematical biology, predator-prey equations, also known as the Lotka-Volterra predator-prey model, describe the population dynamics of two species, one as prey and the other as predator (Lotka, 1925). The equations are given as follows:

$$\frac{dz_1}{dt} = a z_1 - b z_1 z_2, \ \frac{dz_2}{dt} = -c z_2 + d z_1 z_2,$$

where $z_1$ and $z_2$ denote the population of prey and predator, and $\{a, b, c, d\}$ are the parameters of ODE.

**Competition equations.** Competition equations, also known as the Lotka-Volterra competition model, describe the population dynamics among multiple species in competition (Smale, 1976). Under competition among $K$ species, the time derivative of $z_i$, the population of species $i$, is described as follows:

$$\frac{dz_i}{dt} = r_i z_i \left(1 - \sum_{j=1}^K a_{i,j} z_j / \eta_i\right),$$

where $\boldsymbol{r} = \{r_i\}$, $\boldsymbol{\eta} = \{\eta_i\}$, and $\boldsymbol{a} = \{a_{i,j}\}$ for $i = 1, \ldots, K$, $j = 1, \ldots, K$ are the parameters of ODE. $r_i$ denotes the internal growth rate of species $i$. $\eta_i$ denotes the carrying capacity of species $i$. $a_{i,j}$ denotes the degree to which the population of species $j$ decreases the derivative of the population of species $i$. In particular, $a_{i,i}$ is set to 1.

## 2.2 Temporal Point Processes

Temporal point processes (TPPs) are stochastic processes of the event occurrence times. Using TPPs, we can analyze the probabilistic structure of a set of randomly occurring time points, whose nature is determined by a non-negative function, which expresses the rate of events to occur and is called the intensity function.

The Poisson process (Kingman, 1992) is a typical model in which it is assumed that the number of events in the interval $\Delta$ follows a Poisson distribution Poisson $\left(\int_\Delta \lambda(t)dt\right)$, where $\lambda(t)$ is the intensity function. A Poisson process with a time-varying intensity is referred to as an inhomogeneous Poisson process. Furthermore, when the time-varying intensity follows a stochastic process, it is particularly known as a Cox process (Cox, 1955). As a flexible model for fitting the Cox process, Møller et al. developed the log-Gaussian Cox process, in which the time-varying intensity function is modeled as the exponent of a Gaussian process (GP) (Rasmussen & Williams, 2005), which is a distribution over nonlinear functions. To represent the non-negative intensity function by a temporal function $\mathbf{x} \in \mathbb{R}$ generated from the GP, an exponential transformation is used and the intensity at time $t$ is described as follows:

$$\lambda(t) = \lambda_0 \cdot \exp\left(x(t)\right) \ (> 0), \tag{3}$$

where $\lambda_0$ denotes the base intensity, meaning the expected number of events, and $\exp\left(x(t)\right)$ is the modulation.

An approach termed multivariate LGCP introduces correlations between GPs of multiple LGCPs to model interactions among different point processes (Waagepetersen et al., 2016; Hessellund et al., 2022a;b). This is closely related to our study. However, our method differs in that it models the interactions between different point processes using ODEs.

Another major model in the literature on TPPs is a Hawkes process (Hawkes, 1971), a self-excitatory process to model phenomena in which multiple events are concentrated over a relatively short time span. While the proposed model and Hawkes process are similar in that they both have mechanisms that change intensity over time, Hawkes process limits its modeling to self-exciting events. However, the proposed model focuses on a diverse range of discrete events with properties other than self-excitation, through various ODE assumptions.

Recently, Neural TPPs, incorporating the expressive power of neural networks into TPPs, have attracted much attention (Shchur et al., 2021). However, much of the literature on ODE parameter estimation addressed in our study relies on MCMC inference (Gelman et al., 1995; MacKay, 2003). Therefore, we focus on extensions of the classical latent variable type of TPPs (Cox, 1955; Møller et al., 1998) rather than the neural approach, which often has difficulty in MCMC inference (Wenzel et al., 2020; Izmailov et al., 2021).

## 3 Proposed Model

### 3.1 An ODE-guided Poisson Process

We define an ODE-guided Poisson process as a TPP with a time-varying intensity, which follows the dynamics of an ODE. Suppose the ODE consists of $K$ equations with parameters $\boldsymbol{\theta}$, and consider that the events of the $k$-th type are generated from a $k$-th inhomogeneous Poisson process with intensity function $\lambda_k(t)$. Based on Eq. (2) and Eq. (3), considering $\lambda_k(t)/\lambda_0^{(k)} = \exp(x_k(t)) = z_k(t)$ as the $k$-th state quantity in the ODE, the derivatives of the modulation $\exp(\mathbf{x}(t)) = \{\exp(x_k(t))\}_{k=1}^K$ are written as

$$\frac{\partial \exp(\mathbf{x}(t))}{\partial t} = \mathbf{f}\left(\exp\left(\mathbf{x}(t)\right), \boldsymbol{\theta}\right). \tag{4}$$

Here, the state of the ODE corresponds to the modulation of TPP. Because the intensity of the TPP is proportional to the state of the ODE, the higher the state value, the more frequent the event. This relationship between the state and event frequency must hold in the ODE. In practice, ODEs, such as those in the population dynamics literature presented in Section 2.1, need to be chosen or developed based on domain knowledge and assumptions about the phenomena associated with discrete events.

## 3.2 LGCP-based Gradient Matching

Since the inference of ODE-guided Poisson processes requires joint inference of the latent intensities and the parameters of ODE, we propose a new approximate inference framework that combines ODE parameter estimation with LGCP (Møller et al., 1998). The basic approach to ODE parameter estimation, *i.e.*, solving the inverse problem, is numerical integration (Tarantola, 2005). However, this approach is practically inapplicable to complex problems due to computational complexity. Since the scalability of the ODE parameter estimation method with respect to problem complexity is essential for handling a variety of ODEs, we incorporate the gradient matching method, a Bayesian approach originally devised for efficient ODE parameter estimation (Varah, 1982).

In gradient matching, a regression function approximates the trajectory of an ODE by aligning the gradients of the function with the derivatives of the ODE at a series of discrete time points. Inspired by the success of studies on gradient matching using Gaussian processes (Rasmussen & Williams, 2005) as regressors (Calderhead et al., 2008; Dondelinger et al., 2013; Gorbach et al., 2017; Wenk et al., 2019), we modify such GP-based gradient matching (GPGM) to share GPs with LGCPs and develop a gradient matching method applicable to discrete events. The resulting inference framework can be regarded as an extension of fast Gaussian process-based gradient matching (FGPGM) (Wenk et al., 2019), which is a state-of-the-art gradient matching method. We call this framework log-Gaussian Cox process-based gradient matching (LGCP–GM). In the following, we begin with a description of FGPGM and then derive LGCP–GM by making three modifications to FGPGM.

### 3.2.1 FGPGM (Wenk et al., 2019)

In the context of modeling with ODEs, Eq. (1), which describes the observational model, can be described in probabilistic form, as follows:

$$p(\mathbf{y}|\mathbf{z}, \boldsymbol{\sigma}) = \prod_k \mathcal{N}(\mathbf{y}_k|\mathbf{z}_k, \sigma_k^2 \boldsymbol{I}). \tag{5}$$

Then, we build a standard zero-mean GP prior on each latent state $\mathbf{z}_k$, as follows:

$$p(\mathbf{z}|\boldsymbol{\phi}) = \prod_k \mathcal{N}(\mathbf{z}_k|\mathbf{0}, \boldsymbol{C}_{\boldsymbol{\phi}_k}), \tag{6}$$

where $\boldsymbol{\phi} = \{\boldsymbol{\phi}_k\}_{k=1}^K$, and $\boldsymbol{C}_{\boldsymbol{\phi}_k}$ denotes the kernel matrix of a GP with parameters $\boldsymbol{\phi}_k$. Note that the latent variables $\mathbf{z}$ representing the states of the ODE are treated in GPGM as real variables, a setting in which $\mathbf{z}$ can take negative values even in ODEs defined with non-negative states.

Suppose the kernel function is differentiable, then the distribution over the derivatives conditioned on the state is given as follows:

$$
\begin{aligned}
p(\dot{\mathbf{z}}|\mathbf{z}, \boldsymbol{\phi}) &= \prod_k p(\dot{\mathbf{z}}_k|\mathbf{z}_k, \boldsymbol{\phi}_k) \\
&= \prod_k \mathcal{N}(\dot{\mathbf{z}}_k|\mathbf{D}_k \mathbf{z}_k, \mathbf{A}_k), \\
\mathbf{D}_k &= \boldsymbol{C}_{\boldsymbol{\phi}_k}^{\prime\top} \boldsymbol{C}_{\boldsymbol{\phi}_k}^{-1}, \\
\mathbf{A}_k &= \boldsymbol{C}_{\boldsymbol{\phi}_k}^{\prime\prime} - \boldsymbol{C}_{\boldsymbol{\phi}_k}^{\prime\top} \boldsymbol{C}_{\boldsymbol{\phi}_k}^{-1} \boldsymbol{C}_{\boldsymbol{\phi}_k}^{\prime},
\end{aligned}
\tag{7}
$$

where $\boldsymbol{C}_{\boldsymbol{\phi}_k}^{\prime\prime}$ is the autocovariance of each state derivative and $\boldsymbol{C}_{\boldsymbol{\phi}_k}^{\prime}$ is the cross-covariance between the state and its derivative.

FGPGM introduces additional latent variables, $\mathbf{f}_{\text{ode}}$ and $\mathbf{f}_{\text{gp}}$, which have different origins, derivatives of ODE and gradients of GPs, respectively, and considers the equivalence between them[1]. First, given the state $\mathbf{z}$ and the ODE parameters $\boldsymbol{\theta}$, $\mathbf{f}_{\text{ode}}$ can be directly calculated by ODE, *i.e.*, $\mathbf{f}_{\text{ode}} = \mathbf{f}(\mathbf{z}, \boldsymbol{\theta})$ and particularly $\mathbf{f}_{\text{ode}}^{(k)} = \mathbf{f}_k(\mathbf{z}, \boldsymbol{\theta})$ for the $k$-th component. This deterministic relationship between $\mathbf{f}_{\text{ode}}$ and $\mathbf{f}(\mathbf{z}, \boldsymbol{\theta})$ is represented

---

[1]In (Wenk et al., 2019), $\mathbf{f}_{\text{ode}}$ and $\mathbf{f}_{\text{gp}}$ are notated as $\mathbf{F_1}$ and $\mathbf{F_2}$, respectively.

using Dirac's delta functions, as follows:

$$
\begin{aligned}
p\left(\mathbf{f}_{\mathrm{ode}}|\mathbf{z}, \boldsymbol{\theta}\right) &= \prod_k p\left(\mathbf{f}_{\mathrm{ode}}^{(k)}|\mathbf{z}, \boldsymbol{\theta}\right) \\
&= \prod_k \delta\left(\mathbf{f}_{\mathrm{ode}}^{(k)} - \mathbf{f}_k(\mathbf{z}, \boldsymbol{\theta})\right).
\end{aligned}
\tag{8}
$$

Second, given the state $\dot{\mathbf{z}}$, the distribution over $\mathbf{f}_{\mathrm{gp}} = \left\{\mathbf{f}_{\mathrm{gp}}^{(k)}\right\}_{k=1}^{K}$ is given with additional noise scaled by $\gamma$, as follows:

$$
p(\mathbf{f}_{\mathrm{gp}}|\dot{\mathbf{z}}, \gamma) = \prod_k \mathcal{N}\left(\mathbf{f}_{\mathrm{gp}}^{(k)}|\dot{\mathbf{z}}_k, \gamma^2 \boldsymbol{I}\right).
\tag{9}
$$

Finally, considering $\mathbf{f}_{\mathrm{ode}}$ and $\mathbf{f}_{\mathrm{gp}}$ match, this equivalence is represented by Dirac's delta function $\delta\left(\mathbf{f}_{\mathrm{ode}} - \mathbf{f}_{\mathrm{gp}}\right)$. Thus, joint density of latent variables and observation $\mathbf{y}$ can be written as

$$
\begin{aligned}
p\left(\mathbf{z}, \dot{\mathbf{z}}, \mathbf{y}, \mathbf{f}_{\mathrm{ode}}, \mathbf{f}_{\mathrm{gp}}, \boldsymbol{\theta}|\boldsymbol{\phi}, \boldsymbol{\sigma}, \gamma\right) &= p(\mathbf{y}|\mathbf{z}, \boldsymbol{\sigma})p(\mathbf{z}|\boldsymbol{\phi})p(\dot{\mathbf{z}}|\mathbf{z}, \boldsymbol{\phi})p(\boldsymbol{\theta}) \\
&\quad \times p\left(\mathbf{f}_{\mathrm{ode}}|\mathbf{z}, \boldsymbol{\theta}\right) p\left(\mathbf{f}_{\mathrm{gp}}|\dot{\mathbf{z}}, \gamma\right) \delta\left(\mathbf{f}_{\mathrm{ode}} - \mathbf{f}_{\mathrm{gp}}\right),
\end{aligned}
\tag{10}
$$

where $p(\boldsymbol{\theta})$ denotes some prior on the parameters of ODE.

Following Theorem 1 of Wenk et al. and Eqs. (5) to (10), the joint posterior density of $\mathbf{z}$ and $\boldsymbol{\theta}$ in GP-based gradient matching is written as

$$
\begin{aligned}
p(\mathbf{z}, \boldsymbol{\theta}|\mathbf{y}, \boldsymbol{\phi}, \gamma, \boldsymbol{\sigma}) &\propto \prod_k \mathcal{N}\left(\mathbf{y}_k|\mathbf{z}_k, \sigma_k^2 \boldsymbol{I}\right) \times \prod_k \mathcal{N}\left(\mathbf{z}_k|\mathbf{0}, \boldsymbol{C}_{\boldsymbol{\phi}_k}\right) \\
&\quad \times p(\boldsymbol{\theta}) \times \prod_k \mathcal{N}(\mathbf{f}_k(\mathbf{z}, \boldsymbol{\theta})|\mathbf{D}_k \mathbf{z}_k, \mathbf{A}_k + \gamma^2 \boldsymbol{I}).
\end{aligned}
\tag{11}
$$

Note that $\dot{\mathbf{z}}$, $\mathbf{f}_{\mathrm{ode}}$, and $\mathbf{f}_{\mathrm{gp}}$ are now marginalized out and do not appear in Eq. (11). In the last Gaussian factors on the right-hand side of Eq. (11), the variances are small, thus inducing the gradients of GP, $\dot{\mathbf{z}}_k$, to match the derivatives of ODE, $\mathbf{f}_k(\mathbf{z}, \boldsymbol{\theta})$. From a different perspective, for each component $k$, the covariance of the gradient of GP, $\mathbf{A}_k$, along with additional noise scaled by $\gamma$, probabilistically models the expected value of the mismatch between the gradients of both GP and ODE, $\mathbf{f}_k(\mathbf{z}, \boldsymbol{\theta}) - \mathbf{D}_k \mathbf{z}_k$, as a multivariate normal distribution. In other words, these variances control the degree to which violations of the deterministic (*i.e.*, no misalignment allowed) constraints of the ODE are permitted.

### 3.2.2 Derivation of LGCP–GM

In this section, we derive LGCP–GM by making three modifications to FGPGM.

**Modification 1: Switching the likelihood.** A continuous Gaussian likelihood is typically used to model the observation noise in ODE modeling methods, including gradient matching. However, a discrete likelihood, such as Poisson distribution, is necessary for TPPs. To bridge this gap, we use the LGCP's hierarchical structure (Møller et al., 1998), which combines Poisson likelihoods and a Gaussian prior. For an approximate inference on LGCP, time is finely discretized , and event data are converted to count data by counting the event occurrences over each interval. Let $m_{k,t}$ and $x_{k,t}$ be the event count and the state of the $k$-th type at the $t$-th discretized time with interval $\Delta$, respectively. The likelihood of the state $\mathbf{x}$ given event counts $\mathbf{m}$ is evaluated using Poisson distributions as follows:

$$
\begin{aligned}
p(\mathbf{m}|\mathbf{x}) &= \prod_k \prod_t p(m_{k,t}|x_{k,t}) \\
&= \prod_k \prod_t \mathrm{Poisson}\left(m_{k,t}|\Delta \cdot \lambda_0 \cdot \exp(x_{k,t})\right).
\end{aligned}
\tag{12}
$$

**Modification 2: Function completion with sparse GPs.** This LGCP requires a fine discretization of time, whereas gradient matching requires that the number of discrete points at which gradient evaluation is performed be kept smaller to reduce the computational complexity. To resolve this discrepancy, we use the inducing variable method for GPs (Quinonero-Candela & Rasmussen, 2005), which is also known as sparse GPs (Snelson & Ghahramani, 2005). While the inducing variable method was originally devised to reduce the computational complexity of the inverse covariance matrix, we use this method to handle many discretized time points. While we set equally spaced and finely discretized $T$ time points $\boldsymbol{\tau} = \{\tau_t\}_{t=1}^{T}$, referred to as observation points in this study, for evaluating Poisson likelihoods of LGCP, we also set coarsely discretized $U(\ll T)$ time points $\boldsymbol{v} = \{v_u\}_{u=1}^{U}$ as inducing points[2]. Let $\mathbf{x}$ be the state at the inducing points and $\hat{\mathbf{x}}$ be the approximated value of the state at the observation points. The distribution of $\hat{\mathbf{x}}$ is given as the posterior predictive distribution of the Gaussian process conditioned on $\mathbf{x}$, with the covariance approximated diagonally, as follows:

$$
\begin{aligned}
p(\hat{\mathbf{x}}|\mathbf{x}, \boldsymbol{\phi}) &= \prod_k \mathcal{N}\left(\hat{\mathbf{x}}_k \,|\, \boldsymbol{\mu}_{\hat{\mathbf{x}}_k}, \boldsymbol{\Sigma}_{\hat{\mathbf{x}}_k}\right), \\
\boldsymbol{\mu}_{\hat{\mathbf{x}}_k} &= \boldsymbol{C}_{\boldsymbol{\tau v}} \boldsymbol{C}_{\boldsymbol{vv}}^{-1} \mathbf{x}_k, \\
\boldsymbol{\Sigma}_{\hat{\mathbf{x}}_k} &= \operatorname{diag}(\boldsymbol{C}_{\boldsymbol{\tau\tau}} - \boldsymbol{C}_{\boldsymbol{\tau v}} \boldsymbol{C}_{\boldsymbol{vv}}^{-1} \boldsymbol{C}_{\boldsymbol{v\tau}}).
\end{aligned}
\tag{13}
$$

where $\boldsymbol{C}_{..}$ denotes the kernel matrix with parameters $\boldsymbol{\phi}_k$ and $\operatorname{diag}(\boldsymbol{v})$ denotes a diagonal matrix with vector $\boldsymbol{v}$ as its diagonal elements. Subsequently, by restricting the gradient evaluation to the inducing points, the fine discretization for LGCP and the coarse discretization for gradient matching become compatible. In this modification, interpolating the states at the observation points is required for the above compatibility, whereas sparse approximation, *i.e.*, the diagonal approximation of the covariance of the conditional GP, is not essential. However, sparse approximation reduces the computational complexity of evaluating large multivariate normal distributions for the states of the observation points, which is suitable for the requirement of a larger number of observation points[3]. Note that as a result of this modification, $\mathbf{x}$ in Eq. (12) is replaced by $\hat{\mathbf{x}}$, as follows:

$$
\begin{aligned}
p(\mathbf{m}|\hat{\mathbf{x}}) &= \prod_k \prod_t p(m_{k,t}|\hat{x}_{k,t}) \\
&= \prod_k \prod_t \operatorname{Poisson}\left(m_{k,t} | \Delta \cdot \lambda_0 \cdot \exp(\hat{x}_{k,t})\right).
\end{aligned}
\tag{14}
$$

**Modification 3: Transformation of the ODE.** According to Eq. (4), the derivatives of the ODE at the inducing points are evaluated in a non-negative real space for $\exp(\mathbf{x})$, whereas the gradient of the GP is evaluated over the entire real space rather than the nonnegative real space. To match both gradients in the same space in the gradient matching scheme, we redefine the derivatives of the ODE as a function of $\mathbf{x}$ instead of $\mathbf{z} = \exp(\mathbf{x})$, as in Eq. (4), as follows:

$$
\frac{\partial \mathbf{x}}{\partial t} = \mathbf{g}(\mathbf{x}, \boldsymbol{\theta}) = \frac{\mathbf{f}\left(\exp(\mathbf{x}), \boldsymbol{\theta}\right)}{\exp(\mathbf{x})}.
\tag{15}
$$

### 3.2.3 Joint Posterior of LGCP–GM

Considering the above, we modify Eq. (11), the unnormalized joint posterior density of GPGM, to obtain that of LGCP–GM as follows:

$$
\begin{aligned}
p(\hat{\mathbf{x}}, \mathbf{x}, \boldsymbol{\theta}|\mathbf{m}, \boldsymbol{\phi}, \gamma) \propto &\prod_k \prod_t p(m_{k,t}|\hat{x}_{k,t}) \times \prod_k \mathcal{N}\left(\hat{\mathbf{x}}_k \,|\, \boldsymbol{\mu}_{\hat{\mathbf{x}}_k}, \boldsymbol{\Sigma}_{\hat{\mathbf{x}}_k}\right) \times \prod_k \mathcal{N}\left(\mathbf{x}_k|\mathbf{0}, \boldsymbol{C}_{\boldsymbol{\phi}_k}\right) \\
&\times p(\boldsymbol{\theta}) \times \prod_k \mathcal{N}\left(\mathbf{g}_k(\mathbf{x}, \boldsymbol{\theta})|\mathbf{D}_k \mathbf{x}_k, \mathbf{A}_k + \gamma^2 \boldsymbol{I}\right).
\end{aligned}
\tag{16}
$$

If the two probability factors in the second line on the right-hand side of Eq. (16) are ignored (*i.e.*, removal of constraints regarding ODEs), LGCP–GM is reduced to LGCP with sparse GPs, as follows:

---

[2]Note that the number of induction points $U$ must not be so small that the GP conditioned on the induction points cannot capture the global pattern of the function.

[3]Although the computational complexity of a $T$-dimensional Gaussian distribution is $\mathcal{O}(T^3)$, when the covariance matrix is a diagonal matrix, the complexity is reduced to $\mathcal{O}(T)$.

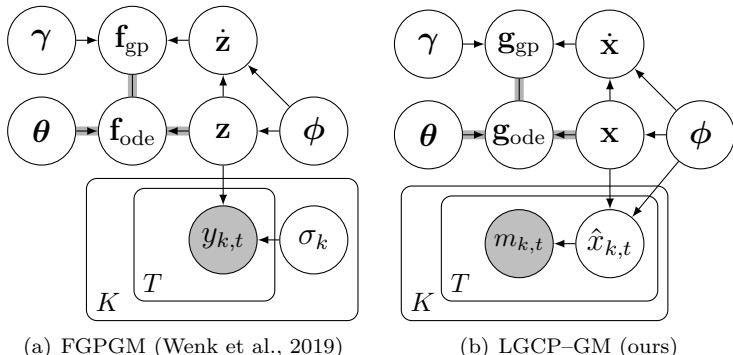

(a) FGPGM (Wenk et al., 2019)  (b) LGCP–GM (ours)

Figure 2: Graphical representations of gradient matching schemes for ODE parameter estimation.

$$p(\hat{\mathbf{x}}, \mathbf{x}|\mathbf{m}, \boldsymbol{\phi}) \propto \prod_k \prod_t p(m_{k,t}|\hat{x}_{k,t}) \times \prod_k \mathcal{N}\left(\hat{\mathbf{x}}_k \mid \boldsymbol{\mu}_{\hat{\mathbf{x}}_k}, \boldsymbol{\Sigma}_{\hat{\mathbf{x}}_k}\right) \times \prod_k \mathcal{N}\left(\mathbf{x}_k|\mathbf{0}, \boldsymbol{C}_{\boldsymbol{\phi}_k}\right). \tag{17}$$

For prior distributions of $\boldsymbol{\theta}$, a logit normal distribution is set as the prior for the scaled variable $s \sim$ Logit$\mathcal{N}(\mu_s, \sigma_s^2)$, where $s = (\theta - \theta_{\min})/(\theta_{\max} - \theta_{\min})$ to set the range $(\theta_{\min}, \theta_{\max})$ on $\boldsymbol{\theta}$ to reflect prior domain knowledge. We treat $\boldsymbol{\phi}$ and $\gamma$ as fixed parameters, as in FGPGM (Wenk et al., 2019). In FGPGM, both $\boldsymbol{\phi}$ and $\boldsymbol{\sigma}$ are set by optimizing GPs on the observed data $\mathbf{y}$, whereas in LGCP–GM, optimizing kernel parameters $\boldsymbol{\phi}$ by applying LGCP to the observed variable $\mathbf{m}$ can easily fall into local optima due to the complexity caused by the non-Gaussian Poisson distribution. Thus, in LGCP–GM, we empirically set the kernel parameters $\boldsymbol{\phi}$ by grid search using the log posterior density evaluated in prior experiments as a metric.

### 3.2.4 Comparison of Graphical Representations

For comparison, Fig. 2 shows graphical models of both the GP-based and LGCP-based gradient matching schemes. There are differences in these graphical models due to the three modifications described above: (1) the shaded node of $m_{k,t}$ and the edge from $\hat{x}_{k,t}$ to $m_{k,t}$ in Fig. 2(b) reflect the first modification regarding Poisson likelihood, (2) the edge from $\boldsymbol{\phi}$ to $\hat{x}_{k,t}$ in Fig. 2(b) reflects the second modification regarding function completion with sparse GPs, and (3) the changes in variables from $\mathbf{f}_{\mathrm{ode}}$ to $\mathbf{g}_{\mathrm{ode}}$ and from $\mathbf{f}_{\mathrm{gp}}$ to $\mathbf{g}_{\mathrm{gp}}$ reflect the third modification regarding the transformation of ODE. Here, $\mathbf{f}_{\mathrm{ode}}$ and $\mathbf{g}_{\mathrm{ode}}$ denote the derivatives of the ODE, as in Eq. (2) and Eq. (15), respectively. In addition, $\mathbf{f}_{\mathrm{gp}}$ and $\mathbf{g}_{\mathrm{gp}}$ denote the gradients of GPs with the addition of a small noise $\boldsymbol{\xi} \sim \mathcal{N}\left(\mathbf{0}, \gamma^2 \boldsymbol{I}\right)$. The shaded edges denote deterministic relationships between variables.

### 3.3 Extrapolation using LGCP–GM

The proposed model allows the estimation of the parameters of ODE within a normalized time $0 \leq t \leq 1$, where data are available, referred to as the interpolation time. A simple extension makes it possible to simultaneously predict the dynamics at the time when no data exist, or in the extrapolation time $1 < t$. Let us introduce inducing points $\boldsymbol{v}'$ and observation points $\boldsymbol{\tau}'$ in extrapolation time and let $x'$ and $\hat{x}'$ be the states at $\boldsymbol{v}'$ and $\boldsymbol{\tau}'$ respectively. $\hat{x}'$ denotes the latent dynamics in the extrapolation time range. Additionally, let $\mathbf{m}'$ be the null count of event occurrence at $\boldsymbol{\tau}'$. By setting $p(\mathbf{m}'|\hat{\mathbf{x}}') = 1$, we can skip the likelihood evaluation for the null counts to avoid erroneous evaluation, as if the event count is not null but zero in the extrapolation time. As a result, we can estimate $\mathbf{x}'$ and $\hat{\mathbf{x}}'$ via the mechanistic knowledge of ODEs learned from discrete event observations at the interpolation time.

## 4 Inference

Because of the non-analytical form of Eq. (16), we require an approximate inference using the Markov chain Monte Carlo method. While previous studies used the random-walk Metropolis-Hastings algorithm

to extract samples from the posterior distribution (Dondelinger et al., 2013; Wenk et al., 2019), we used Hamiltonian Monte Carlo (HMC) method, which is suitable for high-dimensional problems (Duane et al., 1987; Neal, 2011). To use HMC, we need to derive the first derivatives of the log posterior density with respect to each latent variable to be sampled. These can be derived analytically or computed using a library of automatic differentiation, but are not detailed here.

When HMC is applied to a hierarchical Bayesian model, the combination of HMC and blocked Gibbs sampling can be effective (Neal, 2011). In the block sampling strategy (Roberts & Sahu, 1997), the latent variables are partitioned into groups, and a subset of the latent variables in the groups under update is sampled alternately. Therefore, we take the latent variables $\hat{\mathbf{x}}$, $\mathbf{x}$, and $\boldsymbol{\theta}$ as different groups, probabilistically select one or more groups for each iteration of the inference, and repeatedly update the variables in the selected groups.

The high correlation of latent functions, $\mathbf{x}$, that follow GPs makes MCMC inference difficult (Titsias et al., 2011). We use variable transformation to reduce correlations (Kuss et al., 2005). Consider a linear transformation $\mathbf{s} = \mathbf{L}^{-1}\mathbf{x}$, where $\boldsymbol{C}_{\boldsymbol{\phi}} = \mathbf{L}\mathbf{L}^{\top}$ is the Cholesky decomposition. Because $\mathbf{s}$ is white with respect to $\boldsymbol{C}_{\boldsymbol{\phi}}$, we can avoid the high-correlation issue by sampling $\mathbf{s}$ rather than $\mathbf{x}$. Moreover, because the covariance of the GP gradient $\mathbf{A}$ also results in a high correlation of the posterior density, we mitigate the high correlation by approximating $\mathbf{A}$ by its diagonal matrix, $\tilde{\mathbf{A}} = \text{diag}(\mathbf{A})$.

Recall that the last Gaussian factors on the right-hand side of Eq. (16) are multivariate Gaussian distributions with small variances that evaluate the deviation between the gradients of GPs and ODEs. Since the latent variables are randomly initialized at the beginning of the inference, the deviation between the gradients will not be small, and the likelihood of these Gaussian factors will be low. Therefore, in the early iterations of inference, exploration of the state space may occur, such that the likelihood of these Gaussian factors is improved, even as the Poisson likelihoods involving directly observed data are reduced. This exploration may cause a divergence of the state from the region of high probability density in the target distribution, which we originally intend to explore. To address this issue, we use the idea of annealing (Kirkpatrick et al., 1983) to apply the inverse temperature parameter $\beta$ to these Gaussian factors during the burn-in phase of the inference, as follows:

$$
\begin{aligned}
p_{\beta}(\hat{\mathbf{x}}, \mathbf{x}, \boldsymbol{\theta}|\mathbf{m}, \boldsymbol{\phi}, \gamma) \propto & \prod_{k}\prod_{t} p(m_{k,t}|\hat{x}_{k,t}) \times \prod_{k}\mathcal{N}\left(\hat{\mathbf{x}}_{k} \mid \boldsymbol{\mu}_{\hat{\mathbf{x}}_{k}}, \boldsymbol{\Sigma}_{\hat{\mathbf{x}}_{k}}\right) \times \prod_{k}\mathcal{N}\left(\mathbf{x}_{k}|\mathbf{0}, \boldsymbol{C}_{\boldsymbol{\phi}_{k}}\right) \\
& \times p(\boldsymbol{\theta}) \times \prod_{k}\mathcal{N}(\mathbf{g}_{k}(\mathbf{x}, \boldsymbol{\theta})|\mathbf{D}_{k}\mathbf{x}_{k}, \mathbf{A}_{k}+\gamma^2\boldsymbol{I})^{\beta}.
\end{aligned}
\tag{18}
$$

Initially, $\beta = 0$ is used for sampling, and the latent variables are repeatedly updated while gradually approaching $\beta = 1$ in the burn-in iterations. Here, when $\beta = 0$, the Gaussian factors for evaluating the gradient error are completely ignored; when $\beta = 1$, Eq. (18) is consistent with Eq. (16). This allows the inference to begin without diverging the state from the high probability density region of the target distribution.

## 5 Experiments

### 5.1 Settings

This section describes the detailed setup of both the model and the MCMC algorithm. Times were normalized to values between 0 and 1. For observation points, $T = 100$ and $\boldsymbol{\tau} = [0.005, 0.015, \ldots, 0.995]$. For inducing points, $U = 21$ and $\boldsymbol{v} = [0.00, 0.05, \ldots, 1.00]$. We used the squared exponential kernel function with an additive noise term,

$$
k(t, t') = \phi_1^2 \exp\left(-\frac{|t - t'|^2}{2\phi_2^2}\right) + \phi_3^2\delta(t - t'),
$$

where $\phi_1$, $\phi_2$, and $\phi_3$ denote the amplitude, length, and white noise scales, respectively. As mentioned in Section 3.2.3, through preliminary experiments using grid search, we empirically set $\phi_1 = 5.0$, $\phi_2 = 0.15$ for SIR and competition equations and $\phi_2 = 0.1$ for predator-prey equations, $\phi_3 = 0.1$, and $\gamma = 0.1$, as hyperparameters.

Table 1: Root mean squared deviation between the posterior mean of $\boldsymbol{\theta}$ and ground-truth

| Type of ODEs | SIR | | | Predator–Prey | | | Competition | | |
|---|---|---|---|---|---|---|---|---|---|
| base intensity $\lambda_0$ | 50 | 100 | 1000 | 50 | 100 | 1000 | 50 | 100 | 1000 |
| GPGM (T=20) | 0.657 | 0.735 | 0.537 | 7.685 | 6.091 | 13.811 | 3.183 | 3.350 | 2.646 |
| GPGM (T=100) | 0.643 | 0.588 | 0.442 | 4.884 | **1.863** | 2.872 | 3.345 | 3.049 | 2.636 |
| LGCP–GM (ours) | **0.379** | **0.070** | **0.119** | **4.643** | 2.235 | **1.786** | **2.305** | **2.208** | **2.047** |

Table 2: Mean of the negative log-likelihood of posterior samples for extrapolation time ($1 < t \ll 1.5$)

| Type of ODEs | SIR | | | Predator–Prey | | | Competition | | |
|---|---|---|---|---|---|---|---|---|---|
| base intensity $\lambda_0$ | 50 | 100 | 1000 | 50 | 100 | 1000 | 50 | 100 | 1000 |
| LGCP | 619.4 | 464.8 | 5356.7 | 118.1 | 174.7 | 818.5 | 242.6 | 338.2 | 1741.8 |
| LGCP–GM (ours) | **166.5** | **213.2** | **433.2** | **112.8** | **144.1** | **361.7** | **195.4** | **268.6** | **602.3** |

For the HMC setup, the number of leapfrog steps was fixed at 10, and the step size was adjusted within the burn-in MCMC simulations. After $10,000$ burn-in iterations, $1,000$ posterior samples were collected from the subsequent $20,000$ simulations thinned to every 20 samples. Unless an exception is mentioned, these settings were used for the proposed model and baselines in all experiments.

## 5.2 Simulated Study

### 5.2.1 Procedure to Synthesize Experimental Event Data

First, we fixed the initial states and ground-truth parameters of the ODE and simulated the ground-truth dynamics by numerically solving the ODE. Next, we multiplied the simulated states by the same $\lambda_0$ for all $k = 1, \ldots, K$ to obtain the intensity function $\lambda_k(t) = \lambda_0 \exp(z_k(t))$ for the $k$-th component of the ODEs. Then, we discretized the time for $t$ in $[0, 1]$ with an interval of 0.001 and synthesized the data by drawing event counts from Poisson distributions with the parameter $0.001 \times \lambda_k(t)$ at each time point $t$.

### 5.2.2 ODE Parameter Estimation

We compared LGCP–GM with GPGM with respect to the accuracy of ODE parameter estimation. As GPGM, we used slightly modified versions of FGPGM (Wenk et al., 2019) incorporating the inducing variable method and HMC sampling to improve scalability and conduct a fair comparison against LGCP–GM. Since GPGM aggregates event data into bins as count data and models them in discrete time, it is necessary to specify the number of bins for discretization. We used two versions of GPGM as baselines: one with 20 observation points for the coarse discretization case, referred to as GPGM ($T = 20$), and the other with 100 observation points, same as in LGCP-GM, for the fine discretization case, referred to as GPGM ($T = 100$). GPGM ($T = 20$) can be viewed as an alternative baseline for comparison to the original FGPGM, since it has almost the same $\mathbf{y}$ and $\mathbf{z}$ lengths, and its setup is similar to that of the FGPGM with which they coincide. In particular, since GPGM ($T = 100$) has the same number of observation points as LGCP–GM, the choice of likelihood, $i.e.$, Gaussian or Poisson likelihood, is the only difference between it and LGCP–GM. Therefore, GPGM ($T = 100$) is considered a purely comparative benchmark for differences in likelihood. As shown in Fig. 2, GPGM is applied to the observed value of $\mathbf{y}$, which is computed as event counts $\mathbf{m}$ divided by the average number of events per observation point $\lambda_0/T$.

Table 1 shows the root mean squared deviation between the posterior mean of $\boldsymbol{\theta}$ and the ground-truth parameters of ODE as the accuracy metrics of ODE parameter recovery for each method, type of ODEs, and baseline intensity $\lambda_0$, which indicates the rate of data size. The bold font denotes the best result for each case. In most cases, the proposed method outperformed the baseline methods for both the trajectories and parameters of the ODE. In Fig. 3, we draw the estimated ODE trajectories, recovered by numerical

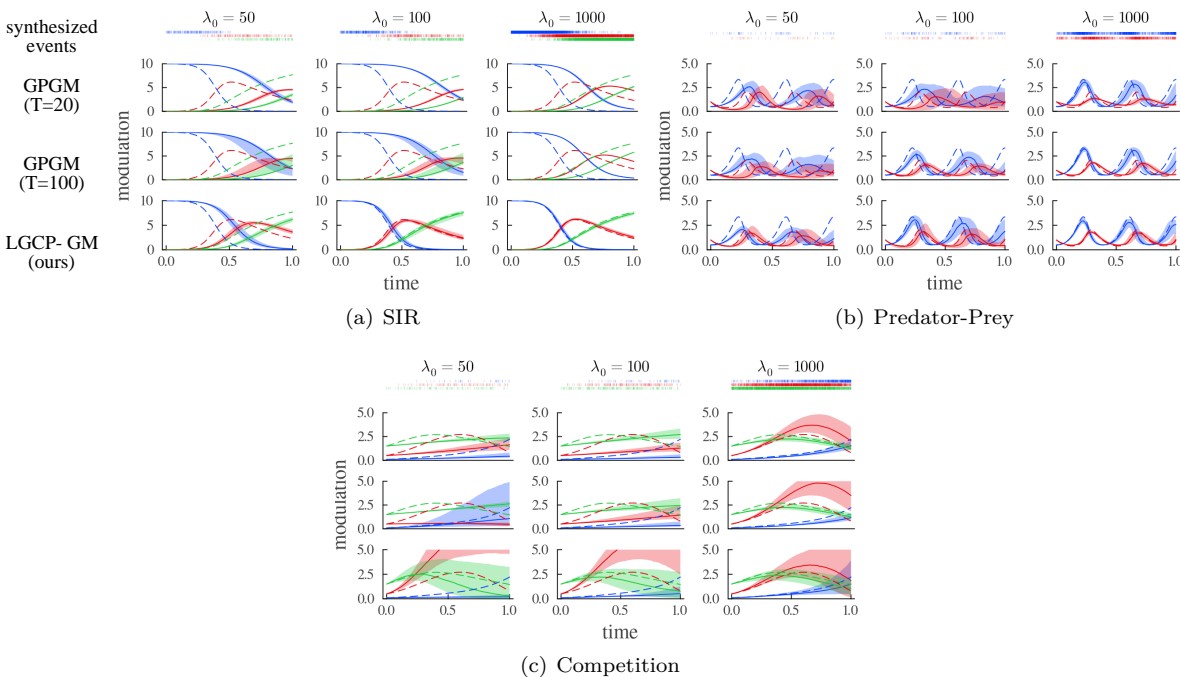

(a) SIR

(b) Predator-Prey

(c) Competition

Figure 3: Numerically simulated modulations for different ODEs and base intensities, using the ground-truth initial state and posterior samples of parameters of ODE. Ground-truth (dashed lines), median (solid lines), and 75% quantiles (shaded areas) of the simulated modulations indicate that LGCP–GM (bottom row) outperforms GPGM baselines with resepect to modulation recovery.

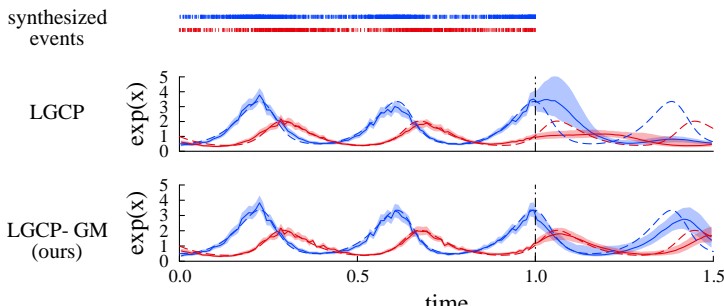

Figure 4: Extrapolation of modulation for predator-prey equations and $\lambda_0 = 1000$. The estimated modulations indicate that LGCP–GM can recover the true modulation (red dashed line) in the extrapolation time (the right side of the dashed-dotted line), but LGCP cannot.

integration using the posterior sample of ODE parameters and the ground-truth initial state, with the synthesized discrete events, and compare them with the ground truth trajectory. For each type of ODE, comparing the results from GPGM with those from LGCP-GM, the recovered ODE trajectories also showed the advantage of the proposed method, especially in settings with little data.

### 5.2.3 Extrapolation

We evaluated extrapolation accuracy for the time $1 < t \ll 1.5$. As a baseline, we used LGCP (Møller et al., 1998) with sparse GPs using Eq. (17) as the posterior density. The kernel parameters $\phi$ for LGCP were determined by a grid search using the log posterior density evaluated in the preliminary MCMC inference as a metric. We used the negative log-likelihood $-\log p(\mathbf{m}'|\hat{\mathbf{x}}')$, as a metric for extrapolation accuracy. To evaluate the negative log-likelihood, we additionally simulated discrete events for the extrapolation time from

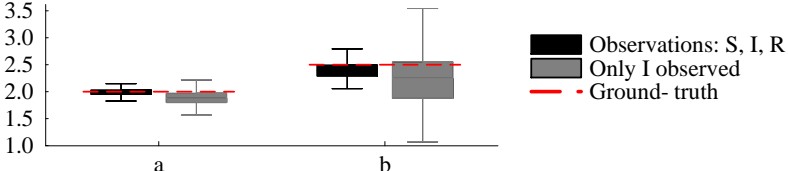

Figure 5: Estimated parameters of SIR model for two cases: (1) all types of events are observed, and (2) only type I events are available.

the ground truth intensities at several different $\lambda_0$ settings and counted the events to calculate $\mathbf{m}'$. This was repeated 100 times to simulate different samples of $\mathbf{m}'$. We evaluated the negative log-likelihood for each combination of simulated $\mathbf{m}'$ and the posterior sample of $\hat{\mathbf{x}}'$ from LGCP and LGCP–GM, respectively.

Table 2 shows the mean of the evaluated negative log-likelihoods for each method, the base intensity, and the type of ODE. The bold font denotes the best result for each case. In all cases, LGCP–GM consistently outperformed LGCP with respect to extrapolation. The performance especially differs in the case with a large base intensity, where the estimated parameters of ODE tend to be more accurate. This suggests that the accurately estimated ODE parameters are useful for extrapolation. Fig. 4 illustrates the estimated modulation for both LGCP and LGCP–GM in the case of predator-prey equations and $\lambda_0 = 1000$. Looking at the ground-truth modulation (red dashed line), median (solid line), and 75% quantile (shaded area) of the estimated modulations, the proposed model succeeded in recovering the dynamics in the extrapolation time, on the right side of the dotted lines, whereas LGCP failed.

### 5.2.4 Modeling on Non-available Events

Consider a scenario where we cannot obtain discrete event observations of certain components in an ODE system consisting of $K$ states. In such a setting, several studies have developed ODE modeling methods for numerical data (Huang et al., 2020; Linial et al., 2021; Yang et al., 2021), and we performed ODE modeling for event data using the proposed method. Let $\mathcal{U}$ be a set of unavailable component indices in the ODE. Based on the same idea of skipping the likelihood evaluation in extrapolation, we can naturally incorporate the unseen $k$-th components of the ODEs by setting the likelihood at observation point $t$ as $p(m_{k,t}|\hat{x}_{k,t}) = 1$ and $p(\hat{x}_{k,t}|\mathbf{x}_k, \sigma_k, \boldsymbol{\phi}_k) = 1$ as long as $k \in \mathcal{U}$. We estimated the parameters of SIR model using only events for I, assuming that events for S and R are unavailable. In addition, we assumed the event count for R at the first observation point to be zero, which softly bound the initial value of the modulation of R to be near zero. This is ad hoc but is a natural assumption in SIR model.

In Fig. 1, we compare two cases: one when events, synthesized with $\lambda_0 = 1000$ for S, I, and R are available, and one when only those for I are available. Fig. 5 shows the estimated parameters for the two cases. Although the posterior variances of both the modulations and parameters of ODE were higher in the second case, the estimation was as successful as in the first case. These results suggest that there are scenarios where the proposed model can be applied, even when some types of events are unavailable.

### 5.3 Applications

**Competition in Technological Industry.** To show an application example of the proposed method to real data, we estimate the competitive relationships among companies related to a specific technological theme using actual patent application data published by the United States Patent and Trademark Office.[4] We extracted patents filed between 2012 and 2020 that had an organization name in the first applicant, had "Handling natural language data" as the International Patent Classification (IPC) label, and included the word "speech" as a keyword in their abstracts. The first applicant (organization) of patents was counted, and the patents of the top five companies in terms of number of patents were extracted. Using the proposed method and competition equations, we analyzed the patent application event series for the 210 patents extracted, as described above. Fig. 6 shows the patent application event data and an adjacency matrix consisting of the estimated competition coefficients. Coefficient $a_{i,j}$ denotes the competitive influence from

---

[4]Data source: USPTO patent application data (https://doi.org/10.7910/DVN/TKURPB)

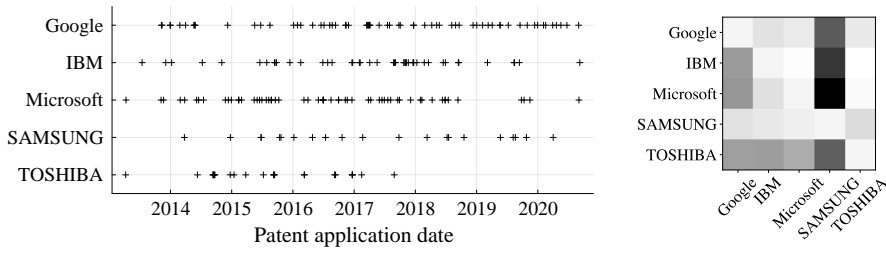

(a) "speech" related patent application events          (b) competition coefficients

Figure 6: (a) Observed events and (b) adjacency matrix consisting of the posterior means of competition coefficients $\boldsymbol{a}$. Darker colors denote a stronger impact of the column component on the row component.

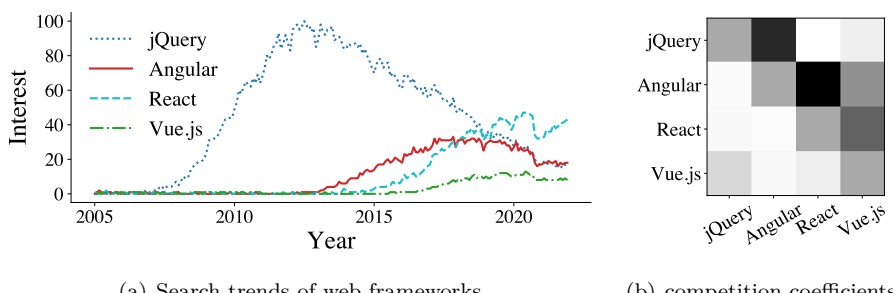

(a) Search trends of web frameworks          (b) competition coefficients

Figure 7: (a) Observed event counts and (b) adjacency matrix in the same style as in Fig. 6(b).

the $j$-th component to the $i$-th component. The row for SAMSUNG has small competition coefficients (indicated by light colors). This corresponds to a steady increase in the number of SAMSUNG's applications, independent of the application trends of other companies. In contrast, the row for TOSHIBA has large competition coefficients (indicated by darker colors). This corresponds to the fact that TOSHIBA's applications decreased during the period when other companies' applications increased; and that it has not filed any applications since 2018. The column for SAMSUNG also has large competition coefficients, corresponding to the fact that the trend of others' applications declined during the period when SAMSUNG's applications increased. Although these results do not directly represent the actual rivalry among these companies, they allow us to estimate the competition coefficients corresponding to changes in patent application trends.

**Competition of Web frameworks.** In the industry of information technology, web application frameworks have several major tools and their popularity has changed over time. Using the proposed model and competition equations, we estimated the competition dynamics between the following representative web frameworks: JQuery, Angular, React, and Vue.js. We retrieved data from Google Trends, a tool that evaluates search volume over time, for worldwide web searches in the Internet and telecommunications category from January 1, 2005, to December 31, 2021. Note that the data obtained with the keyword "Angular" can be a mixture of results for AngularJS and Angular, which are strictly different software. Data consist of positive integers (up to 100) proportional to the search volume, or the string, "less than 1", for each search target and month. We then converted all string values to zeros in order to make all values numeric. We considered these integers as count data aggregated for each month and applied the proposed model. For each component $k$, $\lambda_0^{(k)}$ was set to the total number of counts for events of the $k$-th type. Fig. 7(a) shows the line plots of the data, where the legends are arranged in order of their appearance. Fig. 7(b) shows an adjacency matrix consisting of the estimated competition coefficients, in the same manner as in Fig. 6(b). Looking at the larger competition coefficients represented by darker colors, we can see the impact of Angular on JQuery, React on Angular, and Vue on React. This result illustrates the impact of newer tools on the older ones.

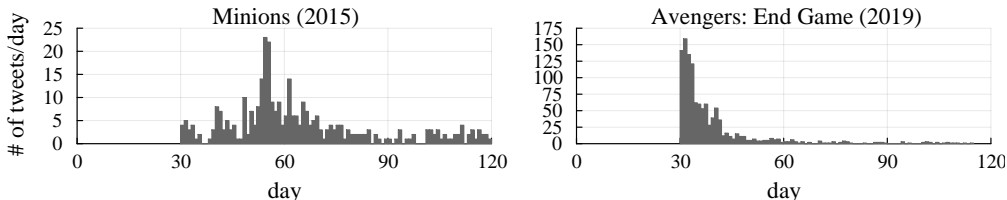

Figure 8: Number of tweets about the two movies over time.

Table 3: Estimated parameters (mean $\pm 2$ standard deviation) of SIR model for two movies

| Movie Production | $a$ | $b$ |
|---|---|---|
| Minions (2015) | $3.03 \pm 0.49$ | $7.13 \pm 1.10$ |
| Avengers: End Game (2019) | $4.46 \pm 0.69$ | $12.98 \pm 1.48$ |

**Trend Spreading on Twitter.** Recently, information sharing among individuals on Twitter has become regular. To understand the trend spreading on Twitter, many attempts have been made to analyze tweet data using epidemiological mathematical models (Abdullah & Wu, 2011; Jin et al., 2013). In these studies, a user who tweeted about a topic was considered as an infectious agent in the mathematical model of infectious diseases with respect to the given topic. Then, using SIR model with S and R as components that cannot be observed, we attempted to analyze movie trends from MovieTweetings (Dooms et al., 2013), a dataset of movie review tweets.[5] Fig. 8 shows the number of tweets per day for 120 days, starting 30 days before the time of the first review tweet for the two films. For all component $k$ including S and R, $\lambda_0^{(k)}$ was set to the total number of tweets. The estimated parameters of SIR model for each movie are presented in Table 3. The magnitudes of $a$ and $b$ can be interpreted as the rates at which the trend rises and then falls. The estimated results show that the first movie generally has a slower rising trend and a more sustained topicality than the second movie. Although this can be observed in Fig. 8, because these movies have significantly different trends, the estimated parameters of ODE allow for numerical comparisons between movies with similar trends and between multiple movies, which is visually difficult.

## 6 Conclusion

Combining temporal point processes with gradient matching methods could increase the accuracy of ODE parameter estimation when the data are irregularly observed discrete events compared with existing gradient matching methods for numerical data, and allow for a suggestive interpretation of the event data via parameters of ODE. Furthermore, exploiting the mechanistic knowledge from the ODEs learned by the proposed method could compensate for the absence of data, making better extrapolation or modeling with unavailable components in the ODE system.

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
