# OpenReview forum: "Estimating Differential Equations from Temporal Point Processes"
_TMLR — Accepted by TMLR_

### Review · Reviewer_5J9q · 2023-07-21

**Summary Of Contributions:**

The paper presents a log-Gaussian Cox process where the intensity is driven by a mechanistic ODE, whose parameters are estimated. The paper adapts a gradient matching based inference towards the model.

**Audience:**

Yes

**Broader Impact Concerns:**

No issues

**Claims And Evidence:**

Yes

**Requested Changes:**

No issues

**Strengths And Weaknesses:**

The paper is well-written and is easy to follow. The method is well derived. The results are sufficient and convincing, and show improvements, which are expected since the underlying mechanistic ODE matches the usecases. The paper has application-focused impact.

The method is incremental, and there is little machine learning novelty. The method derivation also has some issues. The model is not stated in full, but built gradually which makes the final model somewhat implicit. For instance, eq 4 doesn't even define a LGCP, but just a deterministic Cox process. The paper relies heavily on the FGPGM, which also decreases its novelty.

The bigger question I have towards this work is why do we need the gradient matching at all. The paper is mostly about the SIR model, which only has 2 parameters to learn. To fit the SIR to event data, one only then needs to fit 2 parameters and 3 initial states of z's. Surely this is easily doable with off-the-shelf optimisers even with large datasets or long trajectories.

---

> ### Author Response · Authors · 2023-08-02
> **Author response to the reviewer 5J9q**
>
> We thank the reviewer for the thoughtful comments. In our revision, we have made a number of changes that we hope will resolve the reviewer’s concerns (most changes are highlighted in red for your convenience). Below are the answers to the specific questions.
>
> > The bigger question I have towards this work is why do we need the gradient matching at all. The paper is mostly about the SIR model, which only has 2 parameters to learn. To fit the SIR to event data, one only then needs to fit 2 parameters and 3 initial states of z's. Surely this is easily doable with off-the-shelf optimisers even with large datasets or long trajectories.
>
> Actually, the simplicity of the SIR model may allow for parameter estimation without gradient matching. However, with the motivation to broaden the applicability of ODEs, we have examined and argued for the applicability of our method not only to the SIR model but also to other ODEs defined with non-negative states. To claim applicability to various such ODEs, including more complex ones, the parameter estimation method needs to be scalable with respect to the complexity of the model. Therefore, we employ gradient matching.
>
> Based on the above, we have added a note in the first paragraph of section 3.2 that the scalability of the parameter inference method is important for this study. We thank you for your valuable comments.

---

### Review · Reviewer_A6CU · 2023-07-26

**Summary Of Contributions:**

This paper introduces an innovative ODE-based method for concurrently estimating the intensities of multiple point processes. By employing ordinary differential equations controlled by a set of parameters, the model captures the interplay between various point processes, leading to easily interpretable estimates. To estimate intensities in a nonparametric fashion, the approach utilizes Gaussian process regression with MCMC sampling. This method proves beneficial for extrapolation and handling non-available events. The efficacy of the proposed approach is validated through simulation studies and real-world data examples.






**Audience:**

Yes

**Claims And Evidence:**

Yes

**Requested Changes:**

1. Some statements in the paper is inaccurate. For example, one page 1, it is stated that "TPPs model the probability of such event occurrences using a latent intensity function; in particular, a model where the intensity varies over time is called a Cox process (Cox, 1955)." The intensity of a Cox process is not only time-varying but also stochastic (as opposed to a deterministic function). A TPP with a time-varying but deterministic intensity is usually called inhomogeneous/nonhomogeneous Poisson process. Based on my understanding of the paper, the intensities of the point processes under consideration (i.e., exp(x_k(t))) are still deterministic. Therefore, the following term "ODE-guided Cox process" is a bit misleading. I would suggest changing this to "ODE-guided Poisson Process" to be more accurate.

2. The statement in the abstract "our method outperformed existing baseline methods assuming regularly observed continuous
data with respect to the accuracy of recovering the latent parameters of ODEs", is also a bit mis-leading. Based on the likelihood given in (7), the proposed approach also discretize the point process and obtain counts data on a regular grid, which is the same as the baseline method. I think the difference is that the baseline approach assumes a Gaussian likelihood for the counts data and therefore is mis specified. Please clarify this issue.

3. Using ODE to model interactions between different point processes is an interesting idea, but there are other existing approaches for the same purpose. For example, one can assume that each point process is a LGCP and introduce correlations between these latent Gaussian processes, which is commonly referred to as multi-variate LGCP. See, e.g., [1], [2], [3]. The paper seems to completely disregard this line of closely related research. Some comments on the differences between these two approaches are warranted here.

4. For the Gaussian process regression, the choice of the parameters of the covariance kernel function of the latent Gaussian process is important. How do you choose these parameters in your simulation study and real data examples? How sensitive are the performances to the choice of these parameters?


References:

[1] Waagepetersen, R., Guan, Y., Jalilian, A., & Mateu, J. (2016). Analysis of multispecies point patterns by using multivariate log-Gaussian Cox processes. Journal of the Royal Statistical Society Series C: Applied Statistics, 65(1), 77-96.

[2] Hessellund, K. B., Xu, G., Guan, Y., & Waagepetersen, R. (2022). Semiparametric multinomial logistic regression for multivariate point pattern data. Journal of the American Statistical Association, 117(539), 1500-1515.

[3] Hessellund, K. B., Xu, G., Guan, Y., & Waagepetersen, R. (2022). Second-order semi-parametric inference for multivariate log Gaussian Cox processes. Journal of the Royal Statistical Society Series C: Applied Statistics, 71(1), 244-268.




**Strengths And Weaknesses:**

The idea to use ODE to model interactions between different point processes is novel and highly practical.
The paper is well written and the presentation is clear.

---

> ### Author Response · Authors · 2023-08-02
> **Author response to the reviewer A6CU**
>
> We thank the reviewer for the thoughtful comments. In our revision, we have made a number of changes that we hope will resolve the reviewer’s concerns (most changes are highlighted in red for your convenience). Below are the answers to the specific questions.
>
> > The intensity of a Cox process is not only time-varying but also stochastic (as opposed to a deterministic function). A TPP with a time-varying but deterministic intensity is usually called inhomogeneous/nonhomogeneous Poisson process. Based on my understanding of the paper, the intensities of the point processes under consideration (i.e., $\exp(x_k(t))$) are still deterministic. Therefore, the following term "ODE-guided Cox process" is a bit misleading. I would suggest changing this to "ODE-guided Poisson Process" to be more accurate.
>
> Thank you for pointing out the possible misleading points to the reader.
> We agree with your comments and have changed the model name to ODE-guided Poisson process and corrected the related descriptions, including the introduction of inhomogeneous Poisson process, in the revised version.
>
> > The statement in the abstract "our method outperformed existing baseline methods assuming regularly observed continuous data with respect to the accuracy of recovering the latent parameters of ODEs", is also a bit mis-leading. Based on the likelihood given in (7), the proposed approach also discretize the point process and obtain counts data on a regular grid, which is the same as the baseline method. I think the difference is that the baseline approach assumes a Gaussian likelihood for the counts data and therefore is mis specified. Please clarify this issue.
>
> Actually, in LGCP, continuous time is finely discretized and converted into regularly observed count data.
> However, this is an approximate method for handling irregularly observed event data, as is also the case in LGCP-GM.
> Among the baselines, GPGM(T=100) has the same number of observation points as LGCP-GM; therefore, a comparison between LGCP-GM and GPGM(T=100) helps to check the difference in accuracy due to different likelihood choices. This point has been added in section 5.2.2 of the revised version.
>
> > Using ODE to model interactions between different point processes is an interesting idea, but there are other existing approaches for the same purpose. For example, one can assume that each point process is a LGCP and introduce correlations between these latent Gaussian processes, which is commonly referred to as multi-variate LGCP. See, e.g., [1], [2], [3]. The paper seems to completely disregard this line of closely related research. Some comments on the differences between these two approaches are warranted here.
>
> Thank you for your crucial remarks. The multivariate LGCP literature is very closely related to this study, and it would be helpful to the reader to describe the differences between them and this study.
> We have added comments on multivariate LGCP in section 2.2 of the revised version and added citations to relevant studies.
>
> > For the Gaussian process regression, the choice of the parameters of the covariance kernel function of the latent Gaussian process is important. How do you choose these parameters in your simulation study and real data examples? How sensitive are the performances to the choice of these parameters?
>
> As noted in section 3.2, we empirically set the kernel parameters by a grid search using the log posterior density evaluated in prior experiments as a metric.
> The specific appropriate setting depends on the data and the applied ODE model, i.e., the variety of changes in the ODE trajectories.
> However, qualitatively, we empirically found the following behaviors.
>
> In particular, the length scale in LGCP--GM is sensitive to the log-posterior density, a metric in the grid search.
> If the length scale is too short, the GP will not be able to capture the global function pattern, and the variance of the gradient will increase, which will also allow for a larger gradient error in gradient matching.
> Therefore, we can notice this misconfiguration by monitoring the Gaussian likelihood, which evaluates the gradient error.
> Conversely, a too-long length scale can cause GPs to fail in fitting to the data, resulting in incorrectly learned ODE trajectories that do not reflect the data generation process.
> Thus, we can notice this type of misconfiguration by monitoring the Poisson likelihood containing the data.
>
> The amplitude setting is relatively insensitive to the results unless it is so small that the true modulation cannot be recovered.
>
> Finally, the white noise scale must be set reasonably small; too small a setting increases the highly correlated nature of the posterior density distribution and makes MCMC inference more difficult. We can notice this misconfiguration by monitoring MCMC chain getting stuck.

---

> > ### Comment · Reviewer_A6CU · 2023-08-05
> > **Response to response**
> >
> > I would like to thank the authors for addressing my concerns. I only have a little reservation on the last question and wish to get a clarification. When choosing the length scale of the LGCP-GM model, did you impose a prior distribution for this scale parameter and update this scale parameter through MCMC sampling?

---

> > > ### Author Response · Authors · 2023-08-06
> > > **Response to the reviewer A6CU**
> > >
> > > Thank you for checking the revised version. As explained in section 3.2.3, the length scale was given a priori as a hyperparameter by performing a grid search and choosing among several candidate values. In other words, the length scale was not given a prior distribution and was not updated during MCMC.

---

### Review · Reviewer_db6R · 2023-07-27

**Summary Of Contributions:**

ODE modelling and particularly the estimation of their parameters is usually considered from observed data that is regularly sampled and of continuous nature. Importantly, this work considers cases where the event data are discrete/irregularly spaced. This point leads to use temporal point processes (i.e. Cox processes) to model such discrete events. As a consequence, a new ODE-guided Cox process is presented, where the latent intensity function is assumed to follow the dynamics of an ODE. In practice, the work builds on the previous model of [Wenk et al. 2019], where a system of parametric ODEs is combined with GP regression in order to match the time points to the corresponding state values. This one is extended in three specific ways: 1) A switch of likelihood that now is Poisson instead of Gaussian, 2) the introduction of inducing points and 3) a transformation of the ODE to match to the real space (instead of the non-negative).

**Audience:**

Yes

**Claims And Evidence:**

Yes

**Requested Changes:**

>**C1.** Section 3 should be -- *at least* -- as clear as [Wenk et al. 2019] and self-contained. Including an Algorithm would be also helpful and the main differences should be remarked. In some way, the point around if the current work is an extension of Wenk et al. 2019 or not should be extremely clear for the reader since the beginning.

>**C2.** Motivation, details and consequences of including sparse approximations should be discussed and clarified also in section 3 or 4.

>**C3.** The comparison of the proposed model wrt baseline should be as equal as possible, in the sense of having a similar likelihood model or using discrete/continuous events for each one if needed. Just to perceive if the difference of performance is only produced by this difference in the likelihood function or not.

**Strengths And Weaknesses:**

**Strengths.**

The main strengths of the work are perhaps at the beginning and at the end. I particularly liked the presentation/motivation and clarity around the importance of ODE modelling for discrete events or irregularly spaced data for dynamical systems. In that sense, I remark that the paper was clear and concise -- which makes everything more easy to be understood. On the other hand, the experimental results included are meaningful for me, and the authors did an effort to provide sufficient empirical values to prove the performance of their model. In this way, I think that the paper has a complete "tail" of experiments that matches what would be usually required for a submission of this sort to TMLR. However, this does not mean that the empirical results could not be improved, and I will add some comments later, particularly on the way that the method is compared to other baselines.

**Weaknesses.**

In my opinion, the work might be understood as an extension of the ODE model proposed in [Wenk et al. 2019] for accepting discrete events. My main concerns and fears around the novelty/quality of the proposed method come mainly from the section 3. Using [Wenk et al. 2019] as a reference, one can easily see the similarities between both works, and how the current manuscript inherits both the notation and the structure from the other. However, in the current case, the reasons behind the use of a GP regressor and the elements playing some role in the joint distribution of Eq. 5 are not well explained. I mainly say this bc one can easily follow which distribution plays which role in [Wenk et al. 2019] (for instance, Eq. 3, Eq. 4, Eq. 5, Eq. 6, ...), but this is not happening in the current manuscript and I only understood how the model was working once I went to the mentioned reference. In that sense, I would say that the work should be self-contained, not requiring this sort of support from the main SOTA.

Additionally, what is described at the beginning (pp. 2) as a list of contributions, where the ODE-guided Cox process shines as the main one, later seems to be reduced to the three modifications included in section 3. Moreover, in this direction I am concerned about the modification #2, and how and why the inducing variables fit in the ODE method. This is not clear to me. Some questions around this point might be around how large is the computational cost, where does it come from, how the number of inducing points affect to the method, and if is it worth to make such approximation to the ODE model. I can see that [Wenk et al. 2019] chose different approximations and heuristics, that are not considered or mentioned here. I just want to remark that if the previous part was somehow incomplete in the number of details, the one on sparse GPs is even more limited and difficult to understand. Additionally, just for reference, in the GP literature, a change of likelihood plus a sparse approximation and a linking transformation is generally not considered as a novel advance. For this reason, I think is very important to understand why the current approach is novel and important in the ODE context with discrete events.

Also, I am fine with the experiments, and I see that the performance looks relatively good and improved wrt. to the chosen baseline [Wenk et al. 2019]. However, the lingering question in my mind is that if the LGCP model, (which as far as I understood considers a Gaussian likelihood) is trained on discrete events. In that direction, I would like to know just if it makes sense to run a ODE-GP model which has a continuous likelihood function on discrete data, and if this might cause the performance to be significantly worse than the proposed approach.

**References.**
[Wenk et al. 2019] -- Fast Gaussian process based gradient matching for parameter identification in systems of nonlinear ODEs. *Philippe Wenk, Alkis Gotovos, Stefan Bauer, Nico S. Gorbach, Andreas Krause, Joachim M. Buhmann Proceedings of the Twenty-Second International Conference on Artificial Intelligence and Statistics, PMLR 89:1351-1360, 2019.*

---

> ### Author Response · Authors · 2023-08-02
> **Author response to the reviewer db6R**
>
> We thank the reviewer for the thoughtful comments. In our revision, we have made a number of changes that we hope will resolve the reviewer’s concerns (most changes are highlighted in red for your convenience). Below are the answers to the specific questions.
>
> > C1. Section 3 should be -- at least -- as clear as [Wenk et al. 2019] and self-contained. Including an Algorithm would be also helpful and the main differences should be remarked. In some way, the point around if the current work is an extension of Wenk et al. 2019 or not should be extremely clear for the reader since the beginning.
>
> > However, in the current case, the reasons behind the use of a GP regressor and the elements playing some role in the joint distribution of Eq. 5 are not well explained. I mainly say this bc one can easily follow which distribution plays which role in [Wenk et al. 2019] (for instance, Eq. 3, Eq. 4, Eq. 5, Eq. 6, ...), but this is not happening in the current manuscript and I only understood how the model was working once I went to the mentioned reference. In that sense, I would say that the work should be self-contained, not requiring this sort of support from the main SOTA.
>
> > I can see that [Wenk et al. 2019] chose different approximations and heuristics, that are not considered or mentioned here. I just want to remark that if the previous part was somehow incomplete in the number of details, the one on sparse GPs is even more limited and difficult to understand.
>
> We thank you for your important remarks that will make the paper self-contained and will help the reader understand it.
> We have stated in section 3 of the revised version that LGCP-GM can be regarded as an extension of FGPGM, and we have added more detailed descriptions of FGPGM as a separate subsubsection.
>
> > C2. Motivation, details and consequences of including sparse approximations should be discussed and clarified also in section 3 or 4.
>
> > I am concerned about the modification #2, and how and why the inducing variables fit in the ODE method. This is not clear to me. Some questions around this point might be around how large is the computational cost, where does it come from, how the number of inducing points affect to the method, and if is it worth to make such approximation to the ODE model.
>
> > Additionally, just for reference, in the GP literature, a change of likelihood plus a sparse approximation and a linking transformation is generally not considered as a novel advance. For this reason, I think is very important to understand why the current approach is novel and important in the ODE context with discrete events.
>
> Thank you for your perceptive and important remarks.
> The motivation for the introduction of sparse GPs was explained in the part of modification 2.
> However, based on your remarks, we have reconsidered the need for sparse GPs and found that we need to rewrite our discussion of modification 2 as follows:
>
> In modification 2, interpolating the states at the observation points is required for the compatibility of both fine discretization for TPPs and coarse discretization for gradient matching, whereas sparse approximation, i.e., diagonal approximation of the covariance of the conditional GP, is not essential.
> However, sparse approximation reduces the computational complexity of evaluating large multivariate normal distributions for the states of the observation points, which is suitable for the requirement of a larger number of observation points.
>
> This content has been added in the revised version.
> In addition, an explanation of the reduction of computational complexity and the number of induction points has been added to the footnotes of the revised version.
>
> > However, the lingering question in my mind is that if the LGCP model, (which as far as I understood considers a Gaussian likelihood) is trained on discrete events. In that direction, I would like to know just if it makes sense to run a ODE-GP model which has a continuous likelihood function on discrete data, and if this might cause the performance to be significantly worse than the proposed approach.
>
> LGCP considers Poisson rather than Gaussian likelihoods.
> The baseline GPGMs aggregate the frequency of simulated events in bins and model the count data using Gaussian likelihood. Experiment 5.2.2 shows that this approach is less accurate for parameter estimation than LGCP-GM.
>
> > C3. The comparison of the proposed model wrt baseline should be as equal as possible, in the sense of having a similar likelihood model or using discrete/continuous events for each one if needed. Just to perceive if the difference of performance is only produced by this difference in the likelihood function or not.
>
> Among the baselines, GPGM(T=100) has the same number of observation points as LGCP-GM; therefore, a comparison between LGCP-GM and GPGM(T=100) helps to check the difference in accuracy due to different likelihood choices. This point has been added in section 5.2.2 of the revised version.

---

### Decision · Action_Editors · 2023-09-10

**Recommendation:** Accept with minor revision

**Comment:**

Reviewers commended the paper for its clarity as well as the interesting application studies. They commented the method could be useful for many different applications in event data analysis, and they found the results on non-conventional ML applications are worth publishing at TMLR.

The main concern is about novelty, where reviewers pointed out a missing related work (Wenk et al. 2019). In revision the authors included substantial text regarding discussions of Wenk et al. (2019) which has largely cleared reviewers' concern.

In the revision for camera ready, I would suggest the authors to also include some experiments on the fast GPGM method (Wenk et al. 2019) or at least clarify why the original GPGM method serves as a sufficiently good baseline.

**Audience:**

People woking on time-series data. Specifically people interested in neural ODEs, continuous-time methods, and temporal point processes. Also the non-conventional ML applications could be interesting even to general ML audience.

**Claims And Evidence:**

This paper proposes a new ODE-guided Poisson process to estimate differential equations from event data, where the latent intensity function is assumed to follow the dynamics of an ODE. The main new ideas of the proposed method include 1) using Poisson likelihood 2) using inducing points in temporal point process context, and 3) a transformation of the ODE to match to the real space.

Experiments include a simulation study as well as a few interesting non-conventional ML applications (analysing e.g., competitions in tech industry with patent application data, and analysing trend spreading patterns of Twitter with MovieTweetings dataset).